# Invasive Fungal Diseases in Children with Hematological Malignancies Treated with Therapies That Target Cell Surface Antigens: Monoclonal Antibodies, Immune Checkpoint Inhibitors and CAR T-Cell Therapies

**DOI:** 10.3390/jof7030186

**Published:** 2021-03-05

**Authors:** Ioannis Kyriakidis, Eleni Vasileiou, Claudia Rossig, Emmanuel Roilides, Andreas H. Groll, Athanasios Tragiannidis

**Affiliations:** 1Pediatric and Adolescent Hematology-Oncology Unit, 2nd Department of Pediatrics, Faculty of Health Sciences, School of Medicine, Aristotle University of Thessaloniki, AHEPA Hospital, 54636 Thessaloniki, Greece; kyriakidis@auth.gr (I.K.); eleni.vasileiou@yahoo.com (E.V.); 2Department of Pediatric Hematology and Oncology, University Children’s Hospital Münster, D-48149 Münster, Germany; rossig@ukmuenster.de; 3Infectious Diseases Unit, Basic and Translational Research Unit, Special Unit for Biomedical Research and Education, 3rd Department of Pediatrics, Faculty of Health Sciences, School of Medicine, Aristotle University of Thessaloniki, Hippokration General Hospital, 54642 Thessaloniki, Greece; roilides@auth.gr; 4Center for Bone Marrow Transplantation and Department of Pediatric Hematology and Oncology, Infectious Disease Research Program, University Children’s Hospital Münster, D-48149 Münster, Germany; grollan@ukmuenster.de

**Keywords:** invasive fungal diseases, monoclonal antibodies, immune checkpoint inhibitors, CAR T-cells, hematological malignancies, leukemia, lymphoma, children

## Abstract

Since 1985 when the first agent targeting antigens on the surface of lymphocytes was approved (muromonab-CD3), a multitude of such therapies have been used in children with hematologic malignancies. A detailed literature review until January 2021 was conducted regarding pediatric patient populations treated with agents that target CD2 (alefacept), CD3 (bispecific T-cell engager [BiTE] blinatumomab), CD19 (denintuzumab mafodotin, B43, BiTEs blinatumomab and DT2219ARL, the immunotoxin combotox, and chimeric antigen receptor [CAR] T-cell therapies tisagenlecleucel and axicabtagene ciloleucel), CD20 (rituximab and biosimilars, ^90^Y-ibritumomab tiuxetan, ofatumumab, and obinutuzumab), CD22 (epratuzumab, inotuzumab ozogamicin, moxetumomab pasudotox, BiTE DT2219ARL, and the immunotoxin combotox), CD25 (basiliximab and inolimomab), CD30 (brentuximab vedotin and iratumumab), CD33 (gemtuzumab ozogamicin), CD38 (daratumumab and isatuximab), CD52 (alemtuzumab), CD66b (^90^Y-labelled BW 250/183), CD248 (ontuxizumab) and immune checkpoint inhibitors against CTLA-4 (CD152; abatacept, ipilimumab and tremelimumab) or with PD-1/PD-L1 blockade (CD279/CD274; atezolizumab, avelumab, camrelizumab, durvalumab, nivolumab and pembrolizumab). The aim of this narrative review is to describe treatment-related invasive fungal diseases (IFDs) of each category of agents. IFDs are very common in patients under blinatumomab, inotuzumab ozogamicin, basiliximab, gemtuzumab ozogamicin, alemtuzumab, and tisagenlecleucel and uncommon in patients treated with moxetumomab pasudotox, brentuximab vedotin, abatacept, ipilimumab, pembrolizumab and avelumab. Although this new era of precision medicine shows promising outcomes of targeted therapies in children with leukemia or lymphoma, the results of this review stress the necessity for ongoing surveillance and suggest the need for antifungal prophylaxis in cases where IFDs are very common complications.

## 1. Introduction

Molecular targeted therapy is gaining ground in pediatric cancer treatment, especially after the implementation of next-generation sequencing data in clinical practice, and seems to fulfill the “magic bullet” concept that was put forward by German Nobel laureate Paul Ehrlich back in 1900 [1,2]. Targeted therapy differs from the conventional cytotoxic therapy in its specificity of targeted pathways that can halt the growth and spread of cancer cells rather than killing indiscriminately every rapidly dividing cell. Both categories of targeted therapies i.e., monoclonal antibodies (mAbs) and small-molecule inhibitors, have shown great advances since their first members (rituximab in 1997 and imatinib in 2001, respectively) were approved for the treatment of blood cancer in adults [3]. Substantial experience of their use in children, although not widespread, has been achieved in the last decade. While most of these agents seem to be generally well tolerated, opportunistic infections including IFDs should be considered and promptly prevented and treated. However, assessing the exact contribution to infection rates in children with hematologic malignancies receiving biologic therapies is problematic, as many of them are more or less immunocompromised by default and thus at greater risk of infection. Preceding and concomitant immunosuppressive therapies usually render us uncapable of defining the exact relative risk for IFDs conferred by each drug [4]. Host defenses against fungi rely on the sophisticated interplay between: (i) mucocutaneous barrier integrity; (ii) cells of the innate immune system (e.g., dendritic cells and macrophages) that recognize specific pathogen-associated molecular patterns (PAMPs) and bind fungal cell walls using pattern recognition receptors (PRRs) like C-type lectin receptors (CLRs, e.g., dectin-1 recognizing β-glucan, mannose receptor, melanin sensing C-type Lectin receptor MelLec and CARD9 mediator) and Toll-like receptors (TLRs, e.g., TLR2); (iii) cell-mediated immunity via transduction of signals from RPRs and associated molecules (FcRγ, recognition of chitin by the intracellular receptors TLR9 and NOD2) and by phagocytosis, initiation of killing mechanisms (e.g., production of reactive oxygen species), and development of adaptive immunity—especially by CD4+ T-cells producing IFNγ (Th1) or IL-17 (Th17) that attract innate effector cells such as neutrophils and macrophages [5,6]. Based on these patterns, targeted therapies that interfere with host defense at any of the above stages of immune response suggest a predisposition towards the development of IFDs.

An electronic literature search was carried out using the MEDLINE/PubMed and Cochrane Library search engines until January 2021 aiming to identify IFDs in children with hematologic malignancies treated with targeted therapies. Only English language publications and peer-reviewed papers were selected to conduct this review. Moreover, prevalence of IFDs and need for prophylaxis corresponding to each pharmacologic agent will be discussed. Table 1 categorizes and describes all targeted therapies that have been used in children with hematological malignancies (clinical trial identifiers retrieved from: https://www.clinicaltrials.gov/—accessed on 16 December 2020).

## 2. Targeting Antigens on Lymphoid Cells

A number of different monoclonal antibodies targeting antigens on the surface of lymphocytes have been developed and studied. Table 2 summarizes IFDs in children with hematological malignancies treated with agents that target cell surface antigens.

Anti-CD2. Alefacept (Amevive^®^, Biogen, Cambrigde, MA, USA) is a recombinant DNA dimeric fusion protein that consists of the extracellular CD2-binding portion of the human leukocyte function antigen-3 (LFA-3) linked to the Fc portion of human IgG1. By inhibiting LFA-3/CD2 interaction, alefacept interferes with lymphocyte and antigen-presenting cells (APCs) activation, causes a reduction in subsets of CD2+ T-cells (primarily memory effector subsets CD4+CD45RO+ and CD8+CD45RO+), resulting in a reduction in circulating total CD4+ and CD8+ T-lymphocyte counts, while CD2 is also expressed at low levels on the surface of natural killer (NK) cells and certain bone marrow B-cells. Alefacept has been used in two children with acute graft-versus-host disease (GvHD) and one developed aspergillus sinusitis that was successfully treated with surgery and antifungals [25]. Of note, alefacept is no longer being marketed.

Anti-CD3. Until recently, depletion of T-lymphocytes constituted the cornerstone and end point of prophylaxis and treatment against GvHD, given that T-cells mediate immune responses towards allo-antigens. Muromonab-CD3 (Orthoclone OKT3^®^, Ortho-Biotech Products, LP, Bridgewater, NJ, USA) was the first mAb ever to be approved (back in 1985) but is no longer being marketed due to decreased demand and increased infection rates, especially as regards IFDs and in particular invasive aspergillosis. Muromonab targeted CD3, a T-cell co-receptor involved in the activation of both cytotoxic CD8+ naïve T-cells and CD4+ T-helper naïve cells, explaining the detrimental impact of T-cell depletion on IFD occurrence [4].

Anti-CD19. CD19-directed agents that have been used in pediatric blood cancers include the mAbs denintuzumab mafodotin and B43, the bispecific T-cell engagers (BiTEs) blinatumomab and DT2219ARL, and the immunotoxin combotox. The BiTE blinatumomab is the first of the above agents to be approved beyond off-label use and is indicated as monotherapy for the treatment of children aged one year or older with Philadelphia chromosome negative (Ph−) CD19+ B-precursor acute lymphoblastic leukemia (ALL) which is refractory or in relapse (r/r) after receiving at least two prior therapies or in relapse after receiving prior allogeneic hematopoietic stem cell transplantation (AHSCT) [8]. BiTE DT2219ARL, an immunotoxin consisting of two scFv ligands targeting CD19 and CD22 linked to the first 389 amino acids of diphtheria toxin, seems to bear antineoplastic activity but the respective clinical trials involving children with hematologic cancer have not yet published results [26].

The expression of transmembrane protein CD19 is almost exclusively restricted to B-cells and is found during all phases of B-cell development until differentiation into plasma cells, and also B precursor ALL lymphoblasts. Compared with CD20, it is expressed at earlier development stages of B lymphocytes. CD19 function is linked with the decisions of B-cells to survive, proliferate, differentiate, or die. It mainly acts as an adaptor protein to recruit cytoplasmic signaling proteins to the membrane, while it works within the CD19/CD21 complex to decrease the threshold for B-cell receptor (BCR) signaling pathways. Moreover, anti-CD19 agents (as reported for anti-CD20 agents) seem to affect B-cell-dependent activation of T-cells [27]. CD19 is known to play a pivotal role in host defense against fungi by interfering with the threshold for B-cell activation, the complement cascade, and major histocompatibility complex (MHC) class II mediated signaling (affecting the interaction with antigen-presenting cells). Moreover, depletion of CD19-positive B-cells has been associated with hypogammaglobulinemia and delayed recovery of CD4+ T-helper cells [28,29]. BiTE blinatumomab has bi-specificity for CD3 and CD19 proteins and directs CD3+ cytotoxic T-cells to destroy CD19-expressing B-cells. Low serum IgG levels have been associated with blinatumomab administration (2.6% versus none out of 109 patients undergoing conventional chemotherapy), while the rate of neutropenia seems to be lower (37.8% vs. 57.8% of those under conventional chemotherapy). In the previous study of heavily pretreated adults with B-precursor ALL, 3.7% of patients under blinatumomab were reported with IFDs (six out of 267 with invasive pulmonary aspergillosis (IPA), two with mucor-mycosis, one with *pneumocystis jirovecii* pneumonia (PJP), and one with invasive candidiasis) [30]. Immunoglobulin levels may require years to recover and a relevant study showed that only one out of six patients had returned to normal levels in the two-year follow-up, whereas none of them recovered IgA levels [31]. In a study of adult patients with r/r precursor B-ALL under blinatumomab 5, out of 189 patients, 2.6% died of IFDs: two with *Fusarium* infection, one with invasive *Aspergillus* infection, one with invasive *Candida* infection and one with fungal pneumonia. Interestingly, in the latter study, more than 25% of patients developed febrile neutropenia, which was the most common adverse event of blinatumomab treatment [32]. In the same context, another study in adults receiving blinatumomab for r/r precursor B-ALL documented three deaths (8.3%) due to IFDs (described as fungal pneumonia, *Candida* sepsis and disseminated fungal infection of the brain), while the issue of not using antifungal prophylaxis was raised [33]. Systemic trichosporonosis has been reported in a 31-year-old woman with Ph+ ALL under blinatumomab [34]. Of note, only two IFDs have been reported in the literature in children treated with blinatumomab, although Blincyto^®^’s Summary of Product Characteristics (SPC) refers to febrile neutropenia as a common side effect for children (11.4%) and lists invasive mycoses among the most common adverse reactions (10.6%) [8]. The first case of IFD has not been further specified, but it was fatal (one out of 70 pts; 1.4%); severe febrile neutropenia in that study occurred in 17.1% of children [35]. The second case was a child with mucor-mycosis and is described in Table 2 [14]. For all the aforementioned reasons, and beyond the fact that the risk of neutropenia is relatively low, use of blinatumomab should be chaperoned by PJP prophylaxis and when IgG levels remain low for long periods Ig replacement therapy should be considered in selected patients [27].

In a preclinical study, denintuzumab mafodotin showed single-agent efficacy against eight pediatric precursor B-cell lineage ALL patient-derived xenografts, and results from the corresponding clinical trials are anticipated [36]. Another anti-CD19 mAb named B43 conjugated with the tyrosine kinase inhibitor (TKI) genistein has entered clinical trials in children with r/r ALL and non-Hodgkin’s lymphoma (NHL), but no results were found in the literature search. B43 mAb conjugated with pokeweed antiviral protein (B43-PAP) exhibited clinical activity against relapsed precursor B-ALL in children, and, whereas serious infection emerged in three out of 30 patients (two of whom died), severe neutropenia was observed in 9.6% and lymphopenia in 5.9% of the children [37,38]. A phase I study involving children with r/r B-ALL and the humanized anti-CD19 mAb loncastuximab tesirine (or ADCT-402) was terminated early because of slow accrual (NCT02669264).

Immunotoxin combotox is a 1:1 mixture of anti-CD19 and anti-CD22 IgG1 mAbs that are both coupled to deglycosylated ricin-A chain. Combotox has been used in a phase I study in 17 children with r/r precursor B-ALL: three out of 17 children achieved complete remission, three out of 17 children died during the course of treatment, while no IFDs have been reported [39].

Anti-CD20. CD20-directed agents used in treatment of pediatric hematologic malignancies include: (i) first generation mAbs rituximab and ^90^Y-ibritumomab tiuxetan; (ii) second generation mAb ofatumumab with less immunogenicity and improved efficacy (by inducing cell lysis regardless of the level of CD20 expression and in rituximab-resistant CD20+ cells); and (iii) third generation mAb obinutuzumab, that bears an engineered fragment crystallizable region (Fc) to boost complement-dependent cytotoxicity (CDC) and antibody-dependent cellular cytotoxicity (ADCC) [27]. B-cell lysis is induced by CDC, ADCC, apoptosis and sensitization to chemotherapy. Rituximab treatment leads to rapid (within three–seven days) and profound (≥90%) depletion of pre-B and mature B-cells (excluding plasma cells and B- cell precursors) along with CD3+ CD20+ cells (3–5% of T-cells) that lasts about 12 months, while serum immunoglobulin depletion lasts for five to 12 months and may require substitution in selected patients. Rituximab also leads to a reduced function of B-cells as APCs and, by increasing immature and transitional B-cells, to a dysfunction of CD4+ T-cells and abnormal cytotoxic T-cell-specific responses [4]. Anti-CD20 blockade also affects Th17 cells, which are destined to protect mucosal barriers and contribute to pathogen clearance at mucosal surfaces [40].

Similar immunity impairment is observed in all mAbs of this class of agents with extra toxicity against granulocytes for ^90^Y-ibritumomab tiuxetan that is associated with frequent (reaching 68%) and severe neutropenia [41]. Late-onset neutropenia (one to five months after the end of therapy) occurs in 5–15% of patients treated with rituximab, may last for months and is not clearly linked to increased infection rates. It is thought to be immune-mediated as it resolves in many cases spontaneously and as it does not respond to granulocyte colony-stimulating factor (G-CSF) therapy. Late-onset neutropenia is also seen with ofatumumab and obinutuzumab [42]. Anti-CD20 mAbs also exert their cytotoxic activity through antibody-dependent cellular phagocytosis (ADCP), which is 10-fold more cytotoxic than ADCC, independent from ADCC and, in the case of ofatumumab (but less so of rituximab), ADCP is enhanced in the presence of complement. In a related study, venetoclax, Bruton’s tyrosine kinase or BTK-specific inhibitor acalabrutinib, and phosphoinositide 3-kinase or PI3K-delta inhibitor umbralisib (or TGR-1202) seemed to be the preferred candidates for combination therapy with CD20 mAbs in terms of toxicity [43].

A meta-analysis of patients with CD20+ lymphomas reported that addition of rituximab to standard chemotherapy regimens improved overall response while not increasing either overall infection rates (risk ratio (RR) 1; 95% confidence interval (CI): 0.87–1.14) or infection-related mortality (RR 1.6; 95% CI: 0.68–3.75). Nevertheless, the same meta-analysis concluded that rituximab use was associated more than standard chemotherapy with severe neutropenia (RR 1.07; 95% CI: 1.02–1.12) and leucopenia (RR 1.24; 95% CI: 1.12–1.37) [44]. A Cochrane systematic review of adverse events in patients receiving tumor necrosis factor (TNF) blockers (etanercept, adalimumab, infliximab, golimumab, certolizumab pegol), interleukin (IL)-1 antagonist (anakinra), IL-6 antagonist (tocilizumab), anti-CD28 (abatacept), and anti-B cell (rituximab) therapy in patients with any disease condition except human immunodeficiency disease (HIV/AIDS) detected only 12 randomized controlled trials (RCTs; out of 160) with data for fungal infections: only 25 patients in the biologic group versus 23 patients in the control group had fungal infections. In the same study, rituximab was the biologic agent with the lowest odds for serious infections compared with control treatment (OR 0.26; 95% CI: 0.03–2.16) [45]. In the same context, a meta-analysis in patients with rheumatoid arthritis reported that the overall odds ratio for serious infections under rituximab was not significantly increased (OR 1.45; 95% CI: 0.56–3.74) [46]. Another meta-analysis in patients with lymphoma calculated higher risk for PJP in patients under rituximab-containing regimens (RR 3.65; 95% CI: 1.65–8.07) compared to standard chemotherapy. Incidence of PJP in NHL patients receiving rituximab was estimated at just below 3%, while application of PJP prophylaxis had remarkably favorable results (RR 0.28; 95% CI: 0.09–0.94) [47]. Several recent retrospective studies also demonstrated low incidence of PJP (<5%) in lymphoma patients administered rituximab, suggesting that routine PJP prophylaxis may not be universally employed in these patients [48]. According to consensus documents on the safety of anti-CD20 agents, PJP prophylaxis (trimethoprim/sulfamethoxazole: 5–8 mg/kg/day 3 times/week) should only be considered for patients under concomitant therapy with 0.4 mg/kg or 16 mg/day of prednisone or equivalent for at least four weeks, for patients receiving rituximab regimens every 14 days, and, of course, for those with underlying diseases that predispose to IFDs [27,49]. Of note, a retrospective study of 2875 children receiving rituximab revealed a paradox: 3.7% of children with rituximab use for transplantation were found with IFDs (1.4% with invasive candidiasis, 0.9% with aspergillosis, 0.3% with PJP and 1.1% with unspecified mycoses), while the corresponding percentage for children with hematological malignancies was 4% (1.5%, 1%, 0% and 1.5% for each IFD category, respectively), but this difference was not significant [16]. One more study reported a fungal infection in a child with a hematological malignancy under rituximab and has been added in Table 2 [15].

Approved biosimilars of rituximab (Reditux^®^, Rixathon^®^, Ruxience^®^ and others) seem to have similar infection rates with rituximab, even though data for IFDs are currently lacking [50].

^90^Y-ibritumomab tiuxetan, as stated above, is associated with more severe leukopenia, possibly due to radiation effects on cells surrounding CD20+ targets. Nevertheless, the spectrum of infections is similar to that of rituximab [41]. In a study of adults with relapsed CD20+ follicular lymphoma administered with ibritumomab, one out of nine patients (11.1%) developed IFD by *Conidiobolus*, which responded to itraconazole treatment. Subsequently to radioimmunotherapy, all nine patients developed severe neutropenia, the median neutrophil nadir was at week 5 and lasted for a median of 14 days [51]. Data from children in this field are lacking.

Data on IFDs with ofatumumab are sparse. In a phase III study of adults with chronic lymphocytic leukemia (CLL) treated with ofatumumab, the incidence of severe neutropenia was high (13%) compared to the observation group (8%; *p* = 0.11). The latter results were mainly attributed to prolonged and severe neutropenia (24% vs. 10%; *p* = 0.0001) in the ofatumumab arm [52]. Severe neutropenia was documented in 17% of adults patients with r/r CLL receiving ofatumumab, and one PJP case (0.6%; under no prophylaxis) was recorded [53]. Another study of ofatumumab in adults with CLL reported *Candida glabrata* infection in one out of five patients [54]. In a larger retrospective study of 416 adults with lymphoproliferative malignancies and regimens combining bendamustine and anti-CD20 mAb (either rituximab or ofatumumab), 1.7% of patients developed IFDs: five with PJP, 1 with IFD, while 2.2% and 0.5% were on PJP and antifungal prophylaxis, respectively [55].

Obinutuzumab is a third generation anti-CD20 mAb that has been linked with IFDs in adults, and antifungal prophylaxis should be considered, taking into account both concomitant immunosuppressive therapy and underlying condition. Severe neutropenia has been observed in as high as 34% of CLL patients treated with obinutuzumab [56]. Similarly, a phase III study on rituximab-refractory indolent NHL treated with obinutuzumab recorded an incidence rate of 33% of severe neutropenia, while only two cases presented with IFDs (one with PJP and one with fungal sepsis; incidence 0.5% each) [57]. In a study of adults with previously untreated advanced-stage follicular lymphoma under anti-CD20 agents, the overall infection rates were higher in the obinutuzumab arm (77.3%) compared to the rituximab arm (70%), with this difference significantly greater during maintenance therapy (with severe infections in 19.8% and 15.6% of patients, respectively). In the same study, severe neutropenia was documented in 43.9% of subjects treated with obinutuzumab (versus 37.9% with rituximab) [58]. In addition, a phase I study of obinutuzumab in combination with chemotherapy in CD20+ follicular NHL reported severe neutropenia as a common adverse event with 35.8% and 6.9% of affected patients during the induction and maintenance treatment phase, respectively. Severe infections were equally high in both phases of treatment with obinutuzumab (19.8% in the induction, including 1 case of PJP, versus 20.8% in the maintenance phase) [59]. Rare opportunistic infections have been documented with obinutuzumab use. A report from two adult patients with refractory CLL (40 and 38 years old) under obinutuzumab and PJP prophylaxis (cotrimoxazole and pentamidine, respectively) suggested that IFDs are inevitable when neutropenia is prolonged and severe: the first patient developed an invasive *Candida krusei* infection and PJP, while the second patient developed PJP along with bloodstream *Talaromyces marneffei* infection [60]. *T. marneffei* is far more frequent in patients with human immunodeficiency virus infection and acquired immunodeficiency syndrome (HIV/AIDS) in South East Asian countries, but is occasionally seen in those with cell-mediated immunodeficiencies involving the interleukin-12/interferon-γ (IL-12/IFN-γ) signaling pathway (e.g., in patients bearing mutations in the STAT1 gene). Among four reported cases of *T. marneffei* infection in hematology patients undergoing targeted therapies (obinutuzumab, rituximab, ruxolitinib and sorafenib), only a 44-year-old man with CLL under obinutuzumab did not survive (also developing invasive *Candida glabrata* infection and PJP), despite the fact that all patients received the same combination therapy with amphotericin B and voriconazole [61]. Data of IFDs in children administered with obinutuzumab are lacking.

Third generation anti-CD20 mAbs in clinical trials are ocaratuzumab (modified via Fc mutagenesis) and ublituximab and they are both designed to augment ADCC. Severe neutropenia has been reported in 18.9% of adult patients treated with ublituximab for r/r NHL. Safety profiles of these agents are still pending [62,63].

Anti-CD22. Off-label use of CD22-targeted agents in pediatric hematological malignancies involves the immunotoxin combotox (mentioned above with anti-CD19 agents) along with epratuzumab, inotuzumab ozogamicin and moxetumomab pasudotox. CD22 belongs to the SIGLEC (sialic acid-binding immunoglobulin-type lectins) family of I-type lectins and is a sugar binding transmembrane protein that is predominantly found on mature B-cell surfaces and on up to 90% of B-cell blasts and to a lesser extent on some immature B-cells. CD22 seems to regulate B cell receptor activation and subsequently plays a role in B-cell activation and survival, while it also serves as an adhesion molecule [64].

With regard to epratuzumab, a study on children with r/r precursor B-ALL showed that severe infection rates were as high as 50%, while severe neutropenia was observed in only 26.3% of patients [65]. Likewise, a study on adults with r/r ALL treated with epratuzumab recorded febrile neutropenia in 54.8% and severe infections in 41.9% of patients, half of whom eventually died [66]. High rates of severe infections (34.7%) were also demonstrated by a study of epratuzumab use in adults with recurrent indolent NHL [67]. Moreover, radioimmunotherapy with yttrium-90-labelled epratuzumab tetraxetan (⁹⁰Y-DOTA-epratuzumab) in adults with r/r CD22+ precursor B-ALL was associated with IFDs in 11.8% and with severe infections in 52.9% of patients [68]. Nevertheless, long-term safety of repeated courses of epratuzumab therapy in adults with systemic lupus erythematosus reported serious infections in only 6.9%, indicating that concomitant or prior chemotherapy and underlying disease play a pivotal role in infection rates [69]. In view of epratuzumab’s mechanism of action, which is similar to other B-cell-targeted drugs, and given that ensuing neutropenia is not excessively frequent, therapy with this agent does not seem to significantly increase the risk of infection [70].

Inotuzumab ozogamicin (InO, Besponsa^®^, Pfizer/Wyeth, Philadelphia, PA, USA) is an antibody-drug conjugate directed against CD22. In the phase 3 INO-VATE study comparing InO monotherapy with standard of care (SoC) intensive chemotherapy in adults with r/r precursor B-cell ALL, rates of treatment-related severe neutropenia were similar in both arms (36% versus 37.8%), while febrile neutropenia occurred in significantly less subjects in the InO group (26.8% versus 53.8%) and fungal pneumonia was diagnosed only in the SoC arm (2.1%) [71]. Inotuzumab ozogamicin (Besponsa^®^) SPC characterizes fungal infections as a very common adverse reaction among patients treated for relapsed or refractory precursor B-cell ALL (infection rate 48%; neutropenia 49%; febrile neutropenia 26%; fatal infections 5%) [72]. Combination of InO with low-intensity chemotherapy in older patients with Philadelphia chromosome-negative ALL was shown to be safe and effective first-line therapy option, although it was associated with a high rate of severe infections (92.3%) and neutropenia had a median recovery duration of 16 days [73]. Due to the incidence of prolonged neutropenia, antifungal prophylaxis should be considered during expectedly prolonged periods with low neutrophil counts. Although InO does not have any clinically relevant CYP-mediated interactions, it has been shown to increase QTc, and therefore, monitoring is recommended in case of concomitant azole administration. Antifungal prophylaxis with azoles is strongly recommended with InO, but should not be initiated until at least 24 h upon completion of InO treatment [74].

InO is not yet approved for pediatric patients, and clinical studies are ongoing. Data on safety and efficacy in pediatric patients are scarce, but promising. A cohort of 51 children treated with InO for r/r precursor B-ALL on a compassionate use basis reported severe infections in 21.6%, severe and febrile neutropenia in 11.8% and IFDs in 3.9% of patients (one case of candidemia and one of a pulmonary fungal infection) [75]. In another cohort including children with r/r ALL treated with InO only one IFD was recorded (2%), while severe neutropenia was estimated at 44.9% [76]. A recent phase I study of InO in pediatric r/r ALL reported neutropenia in 56% of treated children, but low severe infection rates (8%; one with lung infection and one with sepsis after AHSCT) [77]. A retrospective study of r/r precursor B-cell ALL in pediatric patients under compassionate use with InO and blinatumomab recorded two severe infections (6.5%) that led to death [78]. Likewise, another retrospective study of children with r/r B-ALL treated with InO reported two events of severe infection (7.4%; one with IFD i.e., mucor-mycosis), that survived following bridging therapy with blinatumomab [19].

Moxetumomab pasudotox (Lumoxiti^®^, AstraZeneca, Cambridge, England, UK) is a recombinant anti-CD22 immunotoxin that has been recently evaluated in pediatric patients with r/r B-ALL and a respective phase I study has reported severe neutropenia in 17% and febrile neutropenia in 16% of treated patients [79]. A respective phase II study in children with r/r B-ALL under moxetumomab pasudotox calculated a 20% rate of neutrophil count decreased and a 23.3% rate of febrile neutropenia [80]. Of note, Lumoxiti^®^ is indicated for the treatment of adult patients with r/r hairy cell leukemia (HCL) who received at least two prior systemic therapies, including treatment with a purine nucleoside analog, and contains a boxed warning for capillary leak syndrome and hemolytic uremic syndrome. In a cohort of 28 adult patients with r/r HCL and moxetumomab administration, only one patient presented with severe infection (3.6%) [81]. The extension of the previous study reported only one patient with severe neutropenia (4.8%), but none with febrile neutropenia or serious infection, corroborating that moxetumomab pasudotox spares T-cells and, due to its short half-life and its more sophisticated mechanism of action, averts prolonged B-cell depletion [82].

Anti-CD25. Basiliximab was the first agent of this category to be approved and it is used for the prophylaxis and treatment of GvHD in the pediatric population. Inolimomab is another mAb targeting CD25 that seeks approval for steroid-resistant (SR) acute GvHD. Nevertheless, a recent phase III study on adults with GvHD under inolimomab showed no significant effect on overall survival at one year compared to usual care, while 34.7% of patients experienced at least one fungal infection [83]. CD25 or interleukin-2 receptor alpha chain is a type I transmembrane protein present on activated T- and B-cells (in response to antigenic challenge), whereas it can also be found on thymocytes, myeloid precursors, and oligodendrocytes. Binding of IL-2 to activated T-cells is a critical signal for T-cell proliferation in allograft rejection [84]. Due to its mechanism of action, basiliximab utilization is not expected to affect significantly IFD rates, but concomitant chemotherapy or immunosuppressive therapy and underlying disease remain the key parameters to define the respective risk.

The overall incidence of severe infections among patients treated with basiliximab or placebo in combination with dual and triple immunosuppressive therapy (as prophylaxis and treatment for acute GvHD) was comparable between the groups (26.1% vs. 24.8%), but infections remained the most common cause of death in both groups (1.3% versus 1.4% for the placebo group) [7]. In a cohort of 29 blood cancer patients that included children after haploidentical hematopoietic stem cell transplantation (haplo-HSCT) with G-CSF mobilized bone marrow plus peripheral blood stem cells grafts without T-cell depletion and GvHD prophylaxis of basiliximab, cyclosporine A, methotrexate (MTX), mycophenolate mofetil (MMF) and rabbit anti-thymocyte globulin (ATG) reported an incidence of fungal infections of 13.8% [85]. In a similar cohort that utilized either basiliximab or ATG as part of the GvHD prophylaxis, probable fungal infection rates were higher for ATG (90% vs. 66.7%; all in antifungal prophylaxis). All patients on basiliximab survived, while 43.8% of those receiving ATG failed to respond [86]. In another cohort involving pediatric patients (mean age: 18 years old), 38 of the 53 leukemia patients that were subjected to G-CSF-primed haplo-HSCT (without ex vivo T-cell depletion) received basiliximab in addition to the standard GVHD prophylaxis regimen with ATG, cyclosporine, MTX, and MMF. The latter study had encouraging results, although two patients (5.3%) died of IFDs [87].

Managing severe SR acute GvHD with basiliximab displayed favorable outcomes in a retrospective study of 34 patients (median age: 13 years old), but occurrence of IFDs was the second most fatal adverse event (26% of deceased patients) [88]. A subsequent study with similar design showed that basiliximab was effective in treating steroid refractory acute GvHD after haplo-HSCT in 53 patients aged 8 to 52 years old, while fungal infections occurred in 20.8% of patients [89]. Similar to the studies mentioned above, a recent study including solely pediatric patients concluded that basiliximab is an effective second-line agent, particularly when the skin is involved. Probable or documented fungal infection in this cohort regarded seven out of 100 patients [90]. Conversely, five out of 10 children who underwent haplo-HSCT with T-cell depletion and were treated with basiliximab along with various other agents for hyperacute steroid refractory GvHD developed invasive aspergillosis and did not survive [91]. IFDs in children with hematologic malignancies who received basiliximab are not reported in detail in the literature and precise information is lacking. One interesting case report described a 12-month-old girl who died of *Candida* sepsis after receiving basiliximab for chronic GvHD after ABO-compatible liver transplantation for multi-system Langerhans cell histiocytosis (LCH) [92]. In general, available treatment options for steroid refractory GvHD in infants and young toddlers are limited and ineffective, and anti-CD25 agents are evidently not in the spotlight of current research [93].

Anti-CD30. Iratumumab (MDX-060), a fully humanized IgG1κ mAb that also belongs to the first-generation agents targeting CD30, is a marker of Reed-Sternberg cells in Hodgkin lymphoma (HL) and of anaplastic large cell lymphoma (ALCL), while it is also expressed in various types of other lymphomas and in embryonal carcinoma. Unfortunately, its development was suspended with no explanation given. A phase I/II study in adults receiving iratumumab for HL and ALCL showed inadequate efficacy; severe infection and pneumonia occurred in 9.7% and febrile neutropenia in 1.4% of treated patients [94,95]. The second-generation anti-CD30 mAb XmAb2513 has an Fc region engineered to have increased binding affinity to Fcγ receptors (FcγRs) leading to improved FcγR-dependent effector cell functions. A phase I study in adults with HL showed limited efficacy and one out of 13 patients (7.7%) died of fungal pneumonia [96].

Brentuximab vedotin (*Bv*, Adcetris^®^, Seattle Genetics, Bothell, Washington, WA, USA) is the only mAb of this category to have had active trials for children with hematological malignancy. Brentuximab vedotin was the first antibody-drug conjugate ever to be approved (in 2011) and is used for the treatment of r/r HL and ALCL. CD30 is a cell membrane protein of the TNF receptor family that is expressed by activated, but not by resting, monocytes, T-, B- and NK-cells, where it seems to regulate apoptosis, the balance between Th1 and Th2 responses and to play a crucial role in memory and effector T-cells generation. Anti-CD30 agents interfere with ADCC and can therefore impair humoral immunity [70]. A phase III trial in adults with r/r HL undergoing autologous stem cell transplantation (ASCT) followed by Bv as consolidation therapy noted no significant increase in severe infection rates (6.6% vs. 5.6% in the placebo group), while severe neutropenia was documented in 29.3% of patients (compared to 10% of those under placebo). All patients received PJP prophylaxis adhering to the Adcetris^®^ SPC, which lists PJP and febrile neutropenia among uncommon adverse reactions and contains a boxed warning for progressive multifocal leukoencephalopathy (PML; from JC virus or human polyomavirus 2) [97]. In a large French cohort study of consolidation Bv therapy in r/r HL, severe neutropenia was manifested in 11% and severe infections in 7% of patients [98]. Bv as consolidation therapy after ASCT in children with early r/r HL is feasible but, unlike adults, this treatment option has displayed some significant toxicities (including severe neutropenia in two out of six patients, i.e., 33.3%) [99]. By contrast, a cohort of five children with the same characteristics and under the same regimen reported no cases of neutropenia or infection [100].

Severe neutropenia seems to be the most common (11.1%) adverse event among children receiving Bv for r/r HL and ALCL [101,102]. Indeed, a phase I dose-escalation study of Bv in r/r CD30+ hematologic malignancies reported severe neutropenia in 9.1% of treated patients, but also raised the issue of neuropathy [103]. Another phase I/II study in a pediatric and young adult population under Bv plus gemcitabine for r/r HL reported high rates of severe neutropenia (88.9%), while severe infections were confirmed in 11.1% and febrile neutropenia in 4.4% of patients, respectively [104]. Substitution of vincristine with Bv in the OEPA-COPDAC chemotherapy regimen for children with HL resulted in tenuous and insignificant decrease in severe neutropenia rates (81.3% versus 81.5%) [105]. Preliminary results from an open-label study of Bv and nivolumab, followed by Bv and bendamustine, in 32 pediatric patients with r/r HL and no AHSCT recorded promising outcomes and zero severe infection incidence [106]. A retrospective study from Italy involving patients aged from 12 to 66 years old treated with Bv for r/r HL reported one case of IPA (1.5%) and only three cases of severe neutropenia (4.6%) [107]. Another retrospective study of 19 children treated with Bv for r/r HL recorded pulmonary infection of any grade in 36.8% of patients [108]. A Japanese cohort of six children receiving Bv for r/r HL and ALCL reported respiratory infections in half of the patients (50%; none of them severe) and severe neutropenia in one third of children under treatment [109].

As demonstrated by phase III clinical trials, Bv is not associated with a high risk of infections compared to controls. Beyond transient dose-dependent neutropenia and rare cases of PJP (0.1–1%), Bv administration alone does not require antifungal prophylaxis according to the recent European Conference on Infections in Leukemia (ECIL) position paper for adult patients [110]. In contrast, Bv consolidation regimen in ASCT recipients is recommended to be accompanied by PJP prophylaxis [70]. In the same perspective, a six-year old boy with ACLC and invasive *Candida albicans* infection received consolidation therapy with Bv after living-donor liver transplantation and did not deteriorate, but survived following treatment with caspofungin [111]. IFDs in adults undergoing Bv therapy have been described in several case reports [112,113]. In addition to its designated use, Bv has been also implemented in pediatric embryonal carcinoma treatment with favorable outcomes [114].

Anti-CD33. Gemtuzumab ozogamicin *(GO)* is another antibody-drug conjugate that has been recently approved for r/r CD33+ AML in pediatric patients. It targets CD33 antigen or SIGLEC-3, a transmembrane receptor expressed normally on cells of myeloid lineage, whereas it is usually found on leukemic myeloid cells, on more than 80% of AML blasts, on monocytes, granulocytes and mast cells, but not on normal precursor hematopoietic cells. Due to its cytotoxic effect on immature myeloid cells, CD33-targeted therapy leads to profound and long-lasting neutropenia and correspondingly, to increased infection incidence [9,70]. IFDs following GO use have been under-reported in the literature but their incidence seems to range between 1.3–1.5% [20,115].

Results from a phase III clinical trial in which GO was co-administered with cytotoxic combination therapies in children with newly diagnosed AML demonstrated rates of infection similar to comparator groups in all phases of treatment (induction, intensification, and later transplantation). The rate of documented infection during first induction course with GO was 35.6%, febrile neutropenia emerged in 31.9%, while severe neutropenia affected 23.7% of children and median time to neutrophil recovery was 30 days. As expected, these rates were higher in subsequent treatment phases [116]. A pilot study on GO plus chemotherapy in 340 children with de novo AML reported infections in 44–82% and febrile neutropenia in 13–31% of patients, depending on the phase of treatment [117]. In a recent study on the safety of GO in r/r AML patients, 30.2% of patients under monotherapy presented with severe infections and 47.6% with febrile neutropenia, while the respective rates for GO combined with chemotherapy were 41.5% and 39.3% [118]. A recent study from Germany on 76 children with r/r AML receiving GO revealed that serious infection or serious febrile neutropenia was present in 69% of the cohort (including one case of IPA) during the first cycle of treatment [119]. An older cohort study of 29 children with r/r CD33+ AML under GO reported severe leukopenia in 48%, severe infection with sepsis in 24% and severe pneumonia in 17% of patients, while one child died of fungal sepsis [120].

A retrospective study of 15 children treated with GO for r/r AML or myelodysplastic syndrome (MDS) detected profound pancytopenia and febrile neutropenia in all cases, while one child died suffering from IPA [21]. Another retrospective study from France reported three cases of febrile neutropenia (25%) in 12 children with r/r AML treated with GO [121]. The same group reported on a cohort of 17 children with r/r AML that was administered with GO plus cytarabine, in which severe febrile neutropenia occurred in 35.3%, severe infections in 29.4% and IFDs in 23.5% of patients (including two cases of invasive aspergillosis, one case of aspergillosis reactivation and one case of *Candida kefyr* septicemia), indicating the importance of concomitant immunosuppressive therapy [122]. Relatively high IFD occurrence (two cases out of 15 i.e., 13.3%, both fatal) was denoted in a cohort of children with r/r AML and compassionate GO administration [123]. Likewise, a similar cohort consisting of 12 children reported one lethal case of invasive aspergillosis (8.3%) [22]. A subsequent study of 30 children with r/r AML treated with GO, who received antifungal prophylaxis, recorded severe neutropenia in 93.3% and severe infection in 40% of patients, but no IFDs [124]. A clinical trial assessed the safety of GO in combination with standard chemotherapy regimens in 45 children with AML or MDS and calculated incidence of febrile neutropenia at 28.9%, whereas incidence of severe infections reached 51.1% [125]. An interesting case of a 14-month-old girl with AML that developed post-induction disseminated candidiasis subsequently underwent two GO courses and achieved AHSCT with no IFD active and with the sole complication of 5-day neutropenia [126].

GO has been utilized before AHSCT in r/r AML, but results were not encouraging; febrile neutropenia followed 36.4% of GO infusions and two out of 12 children died of adenovirus infection [127]. Another pilot study of 12 children with r/r CD33+ AML, who received GO combined with busulfan and cyclophosphamide prior to AHSCT, reported severe infections in half of the patients, while one boy developed an IFD despite prophylaxis [23]. Conversely, a single-center report of eight children with r/r AML that received fludarabine, cytarabine, and fractioned GO before AHSCT documented 11 episodes of febrile neutropenia in 13 treatment courses (84.6%) and six episodes of sepsis (46.2%), but no IFDs [128]. GO administration for consolidation after reduced-intensity conditioning and AHSCT in a pediatric cohort with CD33+ AML noted severe neutropenia in all 14 patients (100%), but no IFDs have been documented (under prophylaxis; one death due to progressive disease and respiratory syncytial virus infection) [129]. In the same context, all six children with r/r AML that had received reduced-intensity AHSCT before GO in a pilot study displayed severe neutropenia and four experienced severe bloodstream bacterial infections [130]. A larger cohort of 59 children with post-consolidation GO treatment in r/r AML and after AHSCT showed that severe neutropenia followed 95% and febrile neutropenia 40% of the GO courses [131].

Anti-CD38. Daratumumab, a novel anti-CD38 mAb approved for r/r multiple myeloma, may be an effective option in the treatment of r/r CD38+ hematological malignancies. CD38 is a glycoprotein, cyclic ADP ribose hydrolase, that exerts its function in cell adhesion, signal transduction, cytokine production and calcium signaling, and is expressed at low levels on the surface of many immune cells (CD4+, CD8+, early T- and B-cells and NK cells), plasma cells and germinal center B-cells [70,132]. Patients with T-ALL have robust CD38 surface expression (remaining stable after multi-agent chemotherapy) and a recent preclinical study of pediatric T-ALL patient-derived xenografts (PDX) found daratumumab to be strikingly efficacious in 14 of the 15 different PDXs [133]. In another preclinical study, treatment with daratumumab eradicated minimal residual disease (MRD) in seven of eight pediatric T-ALL PDXs [134]. Off-label use of daratumumab in individual children has shown promising activity and limited toxicity with no indication of IFD predisposition [135]. Clinical studies in ALL are ongoing. Anti-CD38 mAbs eliminate multiple myeloma targets by mediating CDC, ADCC, ADCP, FCγR-mediated cross-linking–induced apoptosis and nicotinamide adenine dinucleotide (NAD+) depletion [132]. The mechanism of action in ALL has not yet been resolved. Isatuximab, another anti-CD38 mAb, is also in a phase II trial for children with r/r ALL and AML (NCT03860844).

Anti-CD52. Alemtuzumab targets the cell surface glycoprotein CD52, which is expressed in high levels on CD3+ T-cells and CD19+ B-cells, at lower levels on NK cells, monocytes and macrophages, while little or no expression is detected on neutrophils, plasma cells or bone marrow stem cells. Binding of alemtuzumab to CD52 results in ADCC and complement-mediated lysis of CD52+ cells. Alemtuzumab is a medication for the treatment of chronic lymphocytic leukemia (CLL) and multiple sclerosis in adults. Lymphopenia is typically profound after administration of alemtuzumab, reaching a nadir within a month of treatment, and lasts for three to 12 months for affected B-cells and up to three years for suppression of CD4+ and CD8+ cells [4]. In a retrospective study of 182 patients (aged 11–79 years old) treated with alemtuzumab, the incidence of IFDs was 17%: 15 cases with invasive aspergillosis (8.2%), 10 with *Candida* infection (5.5%), four with PJP (2.2%), one case of mucor-mycosis and fusariosis (0.5% each). Only an 11-year old boy treated with alemtuzumab for aplastic anemia developed invasive aspergillosis and eventually recovered. Aspergillosis resulted in a higher mortality rate than any other IFD in this cohort despite prophylaxis with fluconazole [136]. A retrospective study of 19 pediatric patients who received alemtuzumab as a single second- or third-line treatment for acute GvHD after AHSCT reported IFDs in four out of 19 cases (21.1%) [17]. In the same context, a cohort of 14 children administered alemtuzumab pre- and post-AHSCT and under fluconazole prophylaxis reported invasive candidiasis in two out of 14 children (14.3%) [18]. A relevant, but older, retrospective study that utilized alemtuzumab as conditioning treatment prior to allo-HCT in nine subjects with severe aplastic anemia reported three deaths of children attributed to IFDs and mismatched grafts (33.3%; one with PJP) [137].

In addition to the aforementioned studies, the literature review revealed a broad spectrum of IFDs among adults receiving anti-CD52 agents, including cryptococcosis, zygo-mycosis, micro-sporidiosis and *Scedosporium* infection [4,138]. A Cochrane review of alemtuzumab use in CLL patients confirmed a significantly elevated risk for infection (RR 1.32; 95% CI: 1.01–1.74) [139]. A literature review calculated the IFD rate at 21.1% in CLL patients treated with alemtuzumab and at 17.5% for those treated for leukemia and lymphomas [138]. Similarly, a retrospective study of 27 patients under alemtuzumab for lymphoproliferative disorders calculated IFD rates as high as 18.5% (three with IPA, one with cryptococcosis and one with histoplasmosis) [140]. Interestingly, a recent prospective study in 71 pediatric patients with nonmalignant disorders and early alemtuzumab treatment as part of the conditioning regimen before allo-HCT reported IFDs at only 5.6% of the cohort (one early, i.e., within 100 days; and three late IFDs, i.e., more than a year post-transplant, all successfully treated). These data suggest a major role of antifungal prophylaxis and underlying disease in IFDs prevalence among children treated with alemtuzumab [141]. According to ECIL-5 guidelines, all children treated with alemtuzumab should receive PJP prophylaxis for at least six months upon completion of treatment [49]. Another consensus document proposed no discontinuation of prophylaxis until peripheral blood CD4+ T-cell count recovers to more than 200 cells/mL, as alemtuzumab-associated lymphopenia clinically behaves like that seen in patients with HIV infection [27]. Alemtuzumab is associated with very high IFD rates, mainly attributed to profound and prolonged T-cell depletion. ATG also displays high IFD rates (but lower than alemtuzumab), because its action affects immune system in a multitude of ways (T-cell depletion, modulation of leukocyte/endothelium interactions, B-cells apoptosis, dysfunction of dendritic cells, and induction of regulatory and natural killer T-cells), but not detrimentally. IFD rates related to basiliximab use are also considerable, but significantly lower than the aforementioned agents [4,142].

Anti-CD66b. ^90^Y-labelled *BW 250/183* (a murine IgG1 mAb directed against carcinoembryonic antigen-related cell adhesion molecule 8 or CEACAM8 or CD66b, that is expressed on the cell surface of almost all human granulocytes and their more mature precursors) is being investigated as a form of radioimmunotherapy in children with r/r leukemia before allo-HCT, as prior dosimetry studies with indium-111 labelled anti-CD66 have shown favorable dose distributions (NCT04082286) [143].

Anti-CD248. Ontuxizumab (MORAb-004) is a mAb directed against the C-type lectin transmembrane receptor endosialin (or tumor endothelial marker-1 TEM-1 or CD248), which is commonly found on the surface of mesenchymal cells of tumor microenvironment, including tumor endothelium, tumor-associated pericytes and activated fibroblasts, which are thought to play a key role in the development of tumor neovascular networks, cell–cell adhesion, stromal interaction and host defense. A phase I study on children with r/r solid tumors showed that ontuxizumab seems to be devoid of infectious complications in that population [144]. A similar study on adults reported only two cases of severe infections (5.6%), but no IFDs [145]. Based on its mechanism of action, ontuxizumab itself is not expected to increase infection risk.

## 3. Immune Checkpoint Inhibitors

### 3.1. Targeting CTLA-4

Ipilimumab and tremelimumab are available cytotoxic T-lymphocyte-associated protein 4 (CTLA-4 or CD152) inhibitors, but with limited off-label use in pediatric hematologic malignancies. CTLA-4 blockade is not expected to predispose to IFDs, because it promotes T-cell priming. Interestingly, CTLA-4 gene knock-out mice seem to develop lymphoproliferative and autoimmune disorders, while clinical experience links these mAbs with various immune-related adverse effects (irAEs) including the induction of autoimmune neutropenia [137]. Treatment of irAEs in the latter case requires corticosteroid administration, in which cases PJP prophylaxis is strongly recommended [138]. Of note, cancer immunotherapy with check point inhibitors can cause irAEs due to loss of regulatory T-cells (Treg) homeostasis [139]. A retrospective study of 740 patients (aged 4–93 years old) treated with immune checkpoint blockers for melanoma showed that pembrolizumab was the safest in terms of serious infection occurrence (odds ratio (OR) 0; 95% CI: 0–0.63), followed by nivolumab (OR 0.29; 95% CI: 0.03–1.68). Odds ratio for serious infection with ipilimumab was 1.05 (95% CI: 0.55–1.9), while, remarkably, the combination of ipilimumab with nivolumab was associated with a relevant risk (OR 3.26; 95% CI: 1.7–6.27). IFDs in the ipilimumab study group were 0.9%: two cases of IPA, three PJP cases, and one *Candida* bloodstream infection, while nine out of 54 patients (17%) died of serious infection. No prophylaxis was given, but this study stresses the need for prophylactic measures especially in patients with irAEs that need prednisone or equivalent for at least four weeks. As expected, risk factors for serious infections in patients undergoing immune checkpoint blockade were use of corticosteroids (OR 7.71; 95% CI: 3.71–16.18) and use of infliximab (OR 4.74; 95% CI: 2.27–9.45) [140]. No published data for IFDs in a pediatric population receiving ipilimumab were revealed, but three cases of IFDs (one with IPA and two with PJP; patient with IPA deceased) were documented in adults, sharing in common the manifestation of irAEs followed by the equivalent immunosuppressive treatment [141,142]. IFD reports are lacking for patients under tremelimumab, but oral candidiasis seems to be a common side effect (5%) [143].

### 3.2. Targeting PD-1 or PD-L1

There are ongoing clinical trials involving children with r/r lymphomas for the agents avelumab, camrelizumab, durvalumab, nivolumab and pembrolizumab, while trials on a pediatric population with atezolizumab have been terminated due to limited activity. Although response to atezolizumab was restricted, it was well-tolerated in children and young adults with previously treated solid tumors, NHL and HL in a phase I/II study (2.2% with severe neutropenia, 1.1% with febrile neutropenia, 13.3% with serial infections) [144]. In general, PD-1/PD-L1 blockade (either as monotherapy or in combination with other anti-cancer drugs) does not seem to increase infection incidence (reporting OR 0.68; 95% CI: 0.67–0.7) [145]. As explained above, increased risk of infection and recommendation for PJP prophylaxis refers only for patients developing irAEs that eventually require additional immunosuppression with corticosteroids and/or TNF-α targeted agents) [138]. Risk of IFDs with TNF-α inhibitors alone or in combination with other immunosuppressive agents (like glucocorticoids) is intrinsically elevated in children [146,147].

PD-1 (programmed cell death protein 1 or CD279) is a transmembrane protein member of the CD28 family on T-, B-, NK and myeloid cells that bears a single extracellular Ig variable domain and a cytoplasmic domain with inhibitory and switch motifs. When PD-L1 (programmed cell death 1 ligand 1 or CD274) binds to PD-1 in the presence of T-cell receptor protein complex (TCR), PD-1 delivers a co-inhibitory signal that terminates TCR/CD28 signal, and thus tumors manage to evade detection and elimination by the immune system. PD-1 activation leads to T-cell anergy and exhaustion, reduced cytotoxicity by NK cells, low cytokine production and inhibitory effects towards myeloid-derived cells like monocytes and macrophages. PD-1/PD-L1 blockade restores T-cell proliferation and the activity of anti-tumor CD8+ T-cells, enhances NK-mediated ADCC, secretes cytokines and attracts APCs [148].

Avelumab is the first mAb of this category to be approved for pediatric patients and is indicated for subjects 12 years and older with metastatic Merkel cell carcinoma. The safety and efficacy of avelumab towards r/r lymphomas is being tested in ongoing clinical trials. Nevertheless, data on children are lacking, but IFDs have not yet been reported with avelumab [149].

Camrelizumab has recently entered clinical trials for r/r HL in children. Reports from adult cohorts display no increase in infection rates. A single-arm study noted severe neutropenia in 2.7% of treated patients, one case of severe upper respiratory tract infection, one case of severe urinary tract infection, but no IFD [150]. Addition of low-dose decitabine to camrelizumab in adults with r/r HL increased the incidence of severe leukocytopenia (37.3% versus zero events in the monotherapy arm), but no severe infections were documented in either group [151]. Another single-arm study of camrelizumab plus gemcitabine, vinorelbine and pegylated liposomal doxorubicin in r/r primary mediastinal B-cell lymphoma (PMBCL) showed that severe neutropenia occurred in 18.5% of patients, while the only documented infection was a case of severe pneumonia (3.7%) [152].

Durvalumab is a mAb that blocks the interaction between PD-L1 and PD-1 and has been studied in combination with ibrutinib in patients with r/r follicular lymphoma or diffuse large B-cell lymphoma (DLBCL). Severe neutropenia was documented in 21.3% of patients, but no severe infection or IFD was recorded [153]. Oral candidiasis may occur (2.1%) but is mild according to Imfinzi^®^ SPC, while severe pneumonia is seen in 3.5%, severe upper respiratory tract infection in 0.2% and severe urinary tract infection in 4% of patients under monotherapy. Of note, febrile neutropenia is observed in 6.4% of patients receiving durvalumab in combination with chemotherapy [154].

Treating pediatric r/r lymphoma patients with *nivolumab* has displayed some initial favorable results. In a study of children with HL, 10 out of 11 children were found with PD-L1 expression in more than 50% of tumor cells [155]. Correspondingly, a recent study on children reported PD-1 expression in 19.4% of HL cases and in 18.2% of r/r HL cases, while PD-L1 expression in more than 50% of tumor cells was documented in 67.7% of HL cases and in 72.7% of r/r HL cases, suggesting that comparable responses to PD-1/PD-L1 blockade would be expected in patients undergoing first-, second-, or third-line therapy [156]. An open-label trial of nivolumab in children with r/r solid tumors and lymphomas reported severe neutropenia in 4.7% and febrile neutropenia in 2.4% of cases [157]. Apart from HL, off-label nivolumab has also been administered to children with advanced malignancies with promising results, even though two out of 10 children developed severe pneumonia [158]. Translation of next-generation molecular diagnostics into a biomarker driven treatment strategy is the aim of the INFORM (INdividualized Therapy FOr Relapsed Malignancies in Childhood) program. In this program, children and adolescents with refractory high-risk malignancies will receive nivolumab in combination with entinostat, a class I HDACi (Histone deacetylase inhibitor) [159]. Low infection rates have been confirmed by a relevant systematic review that calculated incidence of severe infections associated with nivolumab treatment at 0.1% (five cases out of 3386 treated patients; no fatal infection; three with respiratory tract infections and two with urinary tract infections) [160]. Reports for IFDs with nivolumab are limited to adult patients. One case report described exacerbation of IPA in a patient with lung cancer under nivolumab, while a retrospective study of lung cancer patients treated with nivolumab calculated infection incidence at 19.2% with two cases of IFDs among them (one IPA and one *Candida albicans* esophagitis) [161,162]. The literature search revealed two patients with lung cancer that developed fatal PJP while undergoing immunosuppressive therapy for nivolumab-associated immune-related pneumonitis [163]. Another adult patient with lung cancer developed nivolumab-induced severe pancytopenia and was diagnosed with fungal pneumonia due to *Fusarium solani* [164]. Of note, nivolumab has shown promising results in fighting IFDs: PD-1 blockade seems to have efficacy against neutropenic IPA and other IFDs in murine models; in addition, a woman with mucor-mycosis recovered after receiving nivolumab and IFN-γ, supporting the hypothesis that modulation of the immune system could potentially treat some forms of sepsis and IFDs [165,166].

Pembrolizumab is the second PD-1 receptor mAb that has been approved for pediatric use and is applicable in children with refractory HL or HL that has relapsed after three or more prior lines of therapy, in children with refractory primary mediastinal B-cell lymphoma (PMBCL) or PMBCL relapsed after two or more prior lines of therapy, in children with unresectable or metastatic, microsatellite instability-high (MSI-H) or mismatch repair deficient solid tumors that have progressed following prior treatment and who have no satisfactory alternative treatment options, and in children with recurrent locally advanced or metastatic Merkel cell carcinoma [167]. A phase I/II trial of pembrolizumab in pediatric patients with advanced melanoma or a PD-L1-positive, advanced, relapsed, or refractory solid tumor or lymphoma reported only five cases of severe neutropenia in 154 treated children (3.2%), whereas 13% of patients experienced severe infection (with one case of invasive *Candida* infection) and 1.3% encountered febrile neutropenia [168]. In a preceding study of 210 adults with r/r HL under pembrolizumab, the incidence of severe neutropenia was 2.4% and one fatal case of sepsis was noted [169]. Another study of 127 adults with pembrolizumab for advanced rare cancers (whose tumors progressed on standard therapies) reported only one case of severe neutropenia, severe infection in 4.7% and severe lung infection in 3.1% of patients [170]. The literature review revealed two cases of allergic bronchopulmonary aspergillosis, one case of allergic fungal rhinosinusitis and one case of IPA in adults administered with pembrolizumab [171,172,173,174]. The only respective IFD in pediatric population has been added to Table 2.

## 4. Chimeric Antigen Receptor (CAR) T-Cells

CAR T-cell recipients are at risk of infections and IFDs mainly due to prolonged leukopenia, depleted B-cells and low immunoglobulin levels. Other risk factors that predispose to severe infections are multiple previous chemotherapy treatment lines (> 3), infusion of high doses of CAR T-cells (2 × 10^7^ cells/Kg) and the presence of cytokine release syndrome (CRS) or CAR-T-cell-related encephalopathy syndrome (CRES). The last two clinical entities require the administration of tocilizumab as anti-inflammatory agent (an IL-6 mAb) and dexamethasone, respectively, which increases the infection risk. IFDs occurrence has been associated with HCT and presence of CRS, while the proportion of patients that are expected to develop IFDs varies from 0–14.3% (1% with standard prophylaxis). Strict recommendation for antifungal prophylaxis applies in limited cases (prior mold infection, ≥3 weeks of neutropenia before and after CAR T-cell therapy, dexamethasone use > 0.1 mg/Kg/day for at least a week) [175,176]. A recent position paper from Spain stated that PJP prophylaxis along with fluconazole should be standard for children under CAR T-cell therapy, while prophylaxis against filamentous fungi (posaconazole, nebulized liposomal amphotericin B or micafungin) should be added when two or more criteria are met: ≥four prior treatment lines, neutropenia prior to the infusion, CAR-T doses > 2 × 10^7^ cells/Kg, previous IFD, tocilizumab and/or steroids use [177]. Tocilizumab use alone is a risk factor for IFDs and patients should be closely monitored [178]. Documented IFDs in children receiving CAR T-cell therapy for hematologic malignancies are presented in Table 3 [179,180,181,182]. Besides tisagenlecleucel that has been already approved for pediatric r/r B-ALL, another approved CAR T-cell therapy, axicabtagene ciloleucel is being developed for children. It has displayed favorable results in adults with DLBCL and demonstrated low incidence (1%) of IFDs [183]. Interestingly, another cohort of adults treated with CD19 CAR-T cells for DLBCL displayed a cumulative incidence of IFD at one year at 4%, stressing the issue for vigilant monitoring for late (>28 days) IFDs [29]. Brexucabtagene autoleucel (Tecartus^®^, Gilead Sciences, Foster City, CA, USA) has been approved for adults with r/r mantle cell lymphoma, but has not entered trials on children.

## 5. Conclusions

In conclusion, targeted therapies are revolutionizing the treatment of pediatric hematologic cancer in the context of personalized precision medicine, even if they are still used mostly as second- or third- line treatments after the well-established standard of care regimens. As described in detail above, larger datasets on IFDs in children with blood cancer receiving these therapies are lacking, so more retrospective and detailed studies are needed to accurately define IFD risk in each case. The latter task is difficult, because children receiving these therapies have usually been treated before with multiple agents and it is almost impossible for clinicians to decipher the corresponding attributable risk of each agent. For example, IFDs are very common in patients under blinatumomab, inotuzumab ozogamicin, basiliximab, gemtuzumab ozogamicin, alemtuzumab, and tisagenlecleucel (Table 4). BiTE blinatumomab is approved for children with r/r B-cell precursor ALL, while inotuzumab ozogamicin has the same off-label use. Alemtuzumab and basiliximab have been used off-label as part of a condition regimen prior to AHSCT and as GvHD treatment, respectively. Gemtuzumab ozogamicin has recently acquired approval for r/r CD33+ AML in pediatric patients and tisagenlecleucel for r/r B-ALL in relapse post-transplant or in second or later relapse. Bearing in mind all these indications above, it is easy to deduce that children who end up in therapies that target surface antigens have endured long periods of chemotherapy and their immune system is surely more vulnerable. Notwithstanding, immune checkpoint inhibitors have shown favorable results with some agents displaying low incidence of IFDs. In this case, studies must be carefully interpreted, as there are not enough pediatric studies and patients to safely evaluate risk of IFDs. Of course, guidelines for antifungal prophylaxis regarding these therapies should be regularly revised to consider newly emerging data in pediatric patients.

## Figures and Tables

**Table 1 jof-07-00186-t001:** Monoclonal antibodies, immune checkpoint inhibitors and CAR T-cell therapies in clinical trials for children with hematological malignancies.

Name	Trade Name	Target	Clinical Study	Clinical Trial Identifier
**Monoclonal Antibodies (mAbs)**				
Alemtuzumab	Lemtrada	Recombinant humanized rat IgG1κ mAb → CD52	Phase II study in relapsed ALL, phase II in lymphomas, ALL or AML during AHSCT, phase II study in hematological malignancies before PBSCT and AHSCT	NCT00089349, NCT00061945, NCT00040846, NCT00118352, NCT00027560, NCT02061800
Avelumab	Bavencio	Fully human mAb → PD-L1	Phase II study in pediatric lymphoma	NCT03451825
B43	N/A	Murine mAb conjugated with genistein → CD19	Phase I study in recurrent ALL or NHL	NCT00004858
Basiliximab	Simulect	Chimeric mAb → CD25	Prevention of organ transplant rejections	NCT02770430, [7]
Blinatumomab	Blincyto	Recombinant mouse bi-specific T-cell engager (BiTE) → CD19, CD3	Ph-negative CD19-positive B-ALL r/r or after AHSCT, phase III study in relapsed B-ALL	NCT02101853, NCT02393859, NCT02187354, [8]
Brentuximab vedotin (or SGN-35)	Adcetris	Recombinant chimeric IgG1 hamster mAb linked to monomethylauristatin E → CD30 → internalization, tubulin polymerization inhibition, M-phase arrest and apoptosis	Phase II study in stage IIB, IIIB or IV, r/r HL, systemic ALCL, phase II study in r/r HL, phase II study in anaplastic large cell lymphoma, phase III study in HL	NCT01920932, NCT01780662, NCT01979536, NCT02166463, NCT02744612
Camrelizumab (or SHR-1210)	AiRuiKa	Humanized IgG4κ mAb → PD-1	Phase II study in r/r HL, phase III study in r/r HL	NCT04026269, NCT04514081, NCT04233294, NCT04510610
Daratumumab	Darzalex	Recombinant human IgG1κ mAb → CD38	Phase II study in r/r leukemia and lymphoma, phase II study in relapsed AML post-AHSCT	NCT03384654, NCT03537599
Denintuzumab mafodotin (or SGN-CD19A)	N/A	Humanized mAb conjugated with monomethyl auristatin F → CD19	Phase I study in r/r B-ALL and lymphomas, phase I study in r/r B-NHL	NCT01786096, NCT01786135
DT2219ARL	N/A	Bispecific ligand-directed diphtheria toxin (BiTE) → CD19 and CD22	Phase I study in r/r CD19+/CD22+ B-ALL and lymphoma	NCT00889408
Durvalumab	Imfinzi	Human IgG1κ mAb → checkpoint inhibitor blocking the interaction of PD-L1 with PD-1	Phase II study in advanced lymphomas	NCT03837899
Epratuzumab	LymphoCide	Humanized IgG1 mAb → CD22	Phase II/III studies in relapsed ALL	NCT00098839, NCT01802814
Gemtuzumab ozogamicin	Mylotarg	Humanized mouse IgG4κ mAb linked to N-acetyl gamma calicheamicin → CD33	Phase II study in APL and recurrent AML, phase III study in AML, phase I study in r/r AML or MDS, phase II study in AML or MDS before PBSCT, phase II study in AML, phase III study in AML	NCT01409161, NCT00070174, NCT00028899, NCT01869803, NCT00372593, NCT00008151, NCT04326439, NCT04293562, [9]
Ibritumomab tiuxetan	Zevalin	Recombinant murine IgG1 mAb conjugated to the chelating agent MX-DTPA and labelled with ^90^Y (alternatively with indium ^111^In) → CD20	Phase I study in r/r lymphomas	NCT00036855
Inolimomab	Leukotac	Murine IgG1 mAb → CD25	Phase III study in steroid refractory acute GVHD after AHSCT	NCT04289103
Inotuzumab ozogamicin	Besponsa	Recombinant humanizedIgG4κ mAb covalently linked to N-acetyl-gamma-calicheamicindimethyl hydrazide → CD22	Phase II study in MRD+ and r/r CD22+ ALL, phase III study in high-risk B-ALL, mixed phenotype leukemia and B-lymphoma, phase III study in ALL	NCT03913559, NCT03094611, NCT01134575, NCT02981628, NCT03959085, NCT04307576
Ipilimumab	Yervoy	Recombinant fully human IgG1κ mAb → CTLA-4	Phase II study in r/r lymphomas, phase I study in r/r lymphoma, phase I study in MDS after AHSCT	NCT02304458, NCT04500548, NCT01445379, NCT00060372, [10]
Moxetumomab pasudotox tdfk (or CAT-8015)	Lumoxiti	Immunotoxin consisted of mouse Fv mAb fragment fused to a 38 kDa fragment of Pseudomonas exotoxin A (PE38) → CD22 → internalization → ADP-ribosylation of elongation factor 2, inhibition of protein synthesis, and apoptotic cell death	Phase I study in ALL and NHL, phase II study in B-ALL	NCT00659425, NCT02227108
Muromonab-CD3	Orthoclone OKT3	Murine IgG2a mAb → CD3	Phase III study in r/r ALL after AHSCT, phase II study in r/r ALL and lymphoma, APL and MDS before AHSCT	NCT01423747, NCT01423500, NCT00005852
Nivolumab	Opdivo	Human IgG4 mAb → PD-1	Phase I/II study in r/r lymphoma	NCT02304458, NCT04500548
Obinutuzumab (or GA101; formerly afutuzumab)	Gazyva, Gazyvaro	Recombinant humanized IgG1 mAb → CD20	Phase II study in r/r CD20+ NHL	NCT02393157
Ofatumumab (or HuMax-CD20 or OMB157)	Arzerra, Kesimpta	Recombinant human IgG1κ mAb → CD20	Phase II study in recurrent lymphoblastic lymphoma, phase II study in ALL in complete remission	NCT03136146, NCT01363128
Ontuxizumab (or MORAb-004)	N/A	Recombinant chimeric humanized IgG1κ mAb → TEM1	Phase I study in r/r lymphoma	NCT01748721
Pembrolizumab (formerly lambrolizumab)	Keytruda	Humanized IgG4κ mAb → PD-1 receptor	Phase I and II studies in r/r lymphoma	NCT03445858, NCT02332668
Rituximab	MabThera, Truxima, Rituxan, Reditux, Zytux, Rixathon, Riximyo	Chimeric IgG1κ mAb → CD20	NHL and CLL, phase II study in B-ALL and lymphoma, phase II study in r/r B-cell lymphoma during PBSCT, phase I study in r/r lymphoma or leukemia or lymphoproliferative disorder related to AHSCT, phase II study in recurrent lymphoblastic lymphoma, phase I study in r/r lymphomas, phase II study in PMBCL, phase II study in r/r B-ALL	NCT00057811, NCT00058461, NCT00867529, NCT00087009, NCT00324779, NCT03136146, NCT00036855, NCT00983944, NCT01700946, [11]
Tocilizumab	Actemra, RoActemra	Humanized IgG1 mAb → IL6R	Pilot Study in CART19-associated CRS in r/r ALL	NCT02906371
Tremelimumab (formerly ticilimumab)	N/A	Fully human IgG2 mAb → CTLA-4	Phase II study in advanced lymphomas	NCT03837899
**Fusion Proteins**				
Abatacept	Orencia	Extracellular domain of CTLA-4 linked to a modified Fc portion of human IgG1 → inhibits the connection of APCs with CD28 receptor on T-cells by binding CD80 and CD86 → inhibits T-cell activation	Phase II study in GvHD prophylaxis after AHSCT, phase III study in preparative regimen before AHSCT in children with r/r leukemia	NCT01012492, NCT03924401, NCT01743131, NCT04380740, NCT02867800, NCT04000698, [12]
**Chimeric antigen receptor (CAR) T-cells**				
BinD19	N/A	Autologous anti-CD19 CAR TCR-zeta/4-1BB-transduced T lymphocytes	Phase II study in r/r ALL and lymphoma	NCT03265106
CAR-T19/CAR-T22	AUTO3	Autologous T-cells transduced with CD19/22 CAR-ζ/4-1BB vector	Phase I study in r/r CD19+/CD22+ ALL and lymphoma, phase II study in r/r ALL	NCT04204161, NCT03289455
Tisagenlecleucel (or CTL019)	Kymriah	Autologous, immuno-cellular cancer therapy encoding an anti-CD19 CAR	B-ALL and diffuse large B-cell lymphoma, phase II study in r/r B-ALL	NCT02435849, [13]
Axicabtagene ciloleucel	Yescarta	Autologous T-cells against CD19 → T-cell activation via CD28 and CD3-zeta	Phase II study in r/r pre-B-ALL or r/r B-cell NHL	NCT02625480

ABL = Abelson proto-oncogene tyrosine kinase; AFP = alpha fetoprotein; AHSCT = Allogeneic Hematopoietic Stem Cell Transplantation; ALCL = anaplastic large cell lymphoma (type of T-cell non-Hodgkin lymphoma); ALK = anaplastic lymphoma kinase; ALL = Acute Lymphoblastic Leukemia; AML = Acute Myeloblastic Leukemia; APC = antigen-presenting cell; APL = acute promyelocytic leukemia; ASCT = autologous stem cell transplantation; CD3 = cluster of differentiation 3 or T3 complex, a T-cell co-receptor; CD19 = cluster of differentiation 19 or B-lymphocyte surface antigen B4 or T-cell surface antigen Leu-12; CD20 = cluster of differentiation 20 or B-lymphocyte surface antigen B1 or MS4A1 membrane spanning 4-domains A1 protein; CD22 = cluster of differentiation 22 on the surface of mature B-cells; CD25 = cluster of differentiation 25 or IL2RA or interleukin 2 receptor alpha chain; CD28 = cluster of differentiation 28 constitutively expressed on naive T-cells, receptor for CD80 (B7.1) and CD86 (B7.2) proteins; CD30 = TNFRSF8 or tumor necrosis factor receptor superfamily member 8 protein; CD38 = cluster of differentiation 38 or ADPRC1 cyclic adenosine diphosphate (ADP) ribose hydrolase; CD52 = cluster of differentiation 52, a glycoprotein on the surface of mature lymphocytes, monocytes, dendritic cells and mature sperm cells; CD80 = cluster of differentiation 80 or B7-1; CD86 = cluster of differentiation 86 or B7-2 on APCs; CDK = cyclin-dependent kinase; C-Kit = KIT proto-oncogene receptor tyrosine kinase or CD117 or SCFR mast/stem cell growth factor receptor; CLL = chronic lymphocytic leukemia; CML = chronic myelogenous leukemia; CRS = Cytokine Release Syndrome; CSF1R = colony-stimulating factor 1 receptor; CTLA-4 = cytotoxic T-lymphocyte-associated protein 4 or CD152; ERK = extracellular signal-regulated kinase; EZH2 = Enhancer of zester homolog 2; FGFR = fibroblast growth factor receptor; FLT3 = FMS-related tyrosine kinase 3; GvHD = graft-versus-host disease; HL = Hodgkin lymphoma; IgG1κ = immunoglobulin G1 kappa; IgG2 = immunoglobulin G2 isotype; IL-2 = interleukin 2; IL6R = interleukin 6 receptor or CD126; JAK = Janus-associated kinase; JMML = Juvenile Myelomonocytic Leukemia; LCH = Langerhans cell histiocytosis; MAPK = mitogen-activated protein kinase; MDS = myelodysplastic syndromes; MRD = Minimal Residual Disease; N/A = not available; mTOR = serine/threonine-specific protein kinase mammalian target of rapamycin; NHL = non-Hodgkin lymphoma; PARP = poly (ADP-ribose) polymerase; Ph = Philadelphia chromosome; PD-1 = programmed cell death protein 1 or CD279; PDGF-R = platelet-derived growth factor receptor; PD-L1 = programmed death-ligand 1 protein; PBSCT = peripheral blood stem cell transplantation; PI3K = phosphoinositide 3-kinase; PMBLC = primary mediastinal B-cell lymphoma; RAF = Rapidly Accelerated Fibrosarcoma serine/threonine-specific protein kinases; ROS1 = c-ros oncogene 1; r/r = relapsed or refractory; SAHA = suberoylanilide hydroxamic acid; SCFR = stem cell factor receptor; Src = proto-oncogene c-Src tyrosine kinase; TEM1 = tumor endothelial marker 1 or endosialin or CD248; TKI = tyrosine kinase inhibitor; TNFα = tumor necrosis factor-alpha; TNFR2 = tumor necrosis factor receptor superfamily member 1B TNFRSF1B or CD120b; TRK = tropomyosin receptor kinase; TSC = tuberous sclerosis; VEGF-A = vascular endothelial growth factor A; VEGF-R = vascular endothelial growth factor receptor.

**Table 2 jof-07-00186-t002:** Invasive fungal diseases (IFDs) in children with hematological malignancies treated with monoclonal antibodies that target cell surface antigens and immune checkpoint inhibitors.

Reference	Targeted Therapy	Subject	Condition	Concomitant Therapy	Prophylaxis	IFD	Outcome
Schober et al., 2020 [14]	Blinatumomab	7-year-old-boy	Relapsed ALL after AHSCT	No	Yes	Fulminant *Rhizomucor pusillus* mucor-mycosis	Deceased
Kiss et al., 2008 [15]	Rituximab	15-year-old girl	Refractory ALL	6-MP + MTX	N/A	Aspergillosis	Deceased
Kavcic et al., 2013 [16]	Rituximab	19/479 pediatric patients	Hematologic malignancy	N/A	N/A	7 with candidiasis, 5 with aspergillosis, and 7 with unspecified mycoses	N/A
Khandelwal et al., 2014 [17]	Alemtuzumab	4/19 patients (median age: 4 years old)	Steroid-refractory acute GvHD	Various agents in various combinations (methylprednisolone, cyclosporine A, MTX, sirolimus, tacrolimus, ATG, cyclophosphamide)	Yes	Central nervous system fungal infection, candidemia, corneal fungal infection, respiratory fungal infection	2/4 deceased
Shah et al., 2007 [18]	Alemtuzumab	2/14 patients (3 to 17.8 years old)	GvHD prevention in AHSCT	MTX + tacrolimus	Yes	Candidiasis	Both recovered
Contreras et al., 2021 [19]	Inotuzumab ozogamicin	1.8-year old boy	r/r B-ALL with hyper-diploidy	Followed MTX + vincristine/dexamethasone. Blinatumomab “bridging” treatment	N/A	Sinus mucor-mycosis	Recovered
Yamada et al., 2013 [20]	Gemtuzumab ozogamicin	2-year-old girl	Relapsed AML (MLL-MLLT10 rearrangement) complicated with HLH	Sorafenib	N/A	Exacerbation of invasive aspergillosis	Deceased
Liu et al., 2018 [21]	Gemtuzumab ozogamicin	N/A	Refractory CD33+ MDS with monosomy7 & del(5q)	FLAG reinduction	N/A	Invasive aspergillosis	Deceased
Reinhardt et al., 2004 [22]	Gemtuzumab ozogamicin	11.8 year-old girl	r/r AML with del(9)	Steroid prophylaxis for infusion-related side effects	N/A	Invasive aspergillosis	Deceased
Satwani et al., 2012 [23]	Gemtuzumab ozogamicin	12-year-old boy	MDS with monosomy 7	Busulfan + cyclophosphamide	Yes	*Malassezia furfur* sepsis	Survived
Si et al., 2020 [24]	Pembrolizumab	18-year-old girl	Refractory PMBCL	No	No	PJP	Survived

6-MP = 6-mercaptopurine; ATG = anti-thymocyte globulin; FLAG = fludarabine/cytarabine/granulocyte-colony stimulating factor (G-CSF); MTX = methotrexate; N/A = not available; PJP = *Pneumocystis jirovecii* pneumonia; PMBCL: primary mediastinal B-cell lymphoma.

**Table 3 jof-07-00186-t003:** Fungal infections associated with CAR T-cells in pediatric population.

Reference	Target	Cohort	Prophylaxis	IFDs
Ghorashian et al., 2019 [179]	CD19 CAT	14 pts with ALL	N/A	Early: 2 cases of chest fungal infection (14.3%)
Vora et al., 2020 [182]	CD19	81 pts with ALL, 1 with mixed leukemia and 1 with B-cell lymphoma	Cotrimoxazole or inhaled pentamidine; other antifungal prophylaxis at physician’s discretion (18.1%)	Early: 1 case of invasive pulmonary *Cunninghamella* infection (1.2%; unknown if pre-existed and exacerbated); Late: none
Maude et al., 2018 [181]	CD19	75 pts with ALL	N/A	Late: 1 fatal systemic mycosis (1.3%)
Pan et al., 2019 [180]	CD22	26 pts with ALL	N/A	Early: Aspergillus pneumonia (probably present prior to enrollment)

Early (≤28 days) and late (>28 days) fungal infections; pts = patients.

**Table 4 jof-07-00186-t004:** Reported overall incidence of IFDs with various targeted therapies.

Targeted Therapy	Overall Incidence of IFDs in Adults	IFD Reports in Children
Blinatumomab	Very common	Yes
Basiliximab	Very common	Yes
Inotuzumab ozogamicin	Very common	Yes
Gemtuzumab ozogamicin	Very common	Yes
Alemtuzumab	Very common	Yes
Tisagenlecleucel	Very common	Yes
Rituximab	Common	Yes
Axicabtagene ciloleucel	Common	No
Durvalumab	Common	No
Tremelimumab	Common	No
Abatacept	Uncommon	No
Ipilimumab	Uncommon	No
Pembrolizumab	Uncommon	Yes
Avelumab	Uncommon	No
Brentuximab vedotin	Uncommon	No
Moxetumomab pasudotox	Uncommon	No

According to FDA SPCs: very common (≥1/10); common (≥1/100 to <1/10); uncommon (≥1/1000 to <1/100).

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
