# Peer review of "Invasive Fungal Diseases in Children with Hematological Malignancies Treated with Therapies That Target Cell Surface Antigens: Monoclonal Antibodies, Immune Checkpoint Inhibitors and CAR T-Cell Therapies"

_jof, 2021, doi:10.3390/jof7030186_

Round 1
Reviewer 1 Report
This is a comprehensive review of treatments that target cell surface antigens with a focus on invasive fungal infections. Data in the pediatric population are limited as shown in Table 2. The review will be useful to both Hematology and Infectious Disease providers. Particularly ID providers will be educated on the mechanism of action and indications for use of these agents. Table 4 provides a nice summary of the risk of IFI in adults and children.
Introduction: The authors mention that it is difficult to assess the contribution of biologic therapies to infection rates in the setting of underlying immunosuppression. As mentioned later in the article, it should be noted that patients may be receiving multiple chemotherapeutic or biologic agents and it is difficult to ascertain the relative contribution of each drug.
Blinatumomab, rituximab: Please elaborate on why B-cell depleting agents increase the risk of invasive fungal infection. Can discuss the complex interplay between different arms of the immune system.
Lines 165-166: Please clarify how anti-CD20 blockade differs from anti-CD19.
Line 295: “In a larger retrospective study of adults” – How many patients were included?
Lines 376-377: The authors probably mean that antifungal prophylaxis with azoles should not start until at least 24 hours after completion of inotuzumab.
Basiliximab: Please include a brief discussion on the rates of IFI in comparison to ATG and alemtuzumab. One would expect higher rates of IFI with T-cell depleting agents.
Lines 611-613: “No IFDs were identified”. Would rephrase: Only an 11-year old boy was identified with invasive aspergillosis.
Immune checkpoint inhibitors: Suggest to include an introduction explaining that the risk of infection is increased in those receiving immunosuppressive therapy (steroids, TNF-alpha blockers) due to immune-mediated adverse events. This is briefly mentioned in the section but would be worth to emphasize.
Author Response
RESPONSE TO REVIEWER #1
Thank you for providing a thorough review and your insightful comments.
Introduction: we added the sentence “Concomitant immunosuppressive therapies usually render us uncapable of defining the exact relative risk for IFDs conferred by each drug” in lines 60-62, so as to underline that it is difficult to ascertain the relative contribution of each chemotherapy agent to IFD occurrence.
As regards anti-CD19 and anti-CD20 blockade we added 3 more sentences to elucidate the immunologic consequences observed in these therapies: “CD19 is known to play a pivotal role in host defense against fungi by interfering with the threshold for B-cell activation, the complement cascade, and major histocompatibility complex (MHC) class II mediated signaling (affecting the interaction with anti-gen-presenting cells). Moreover, depletion of CD19-positive B-cells has been associated with hypogammaglobulinemia and delayed recovery of CD4+ T-helper cells.” […] “Anti-CD20 blockade also affects Th17 cells, which are destined to protect mucosal barriers and contribute to pathogen clearance at mucosal surfaces.” After these addition we feel that the differences between anti-CD20 and anti-CD19 have been adequately described.
The study of Sarlo et al. (2020) had 416 participants and that sample size has been added in text.
The sentence “Delayed antifungal prophylaxis with azoles for 376 at least 24 hours after completion of an InO course is strongly recommended” has been rephrased: “Antifungal prophylaxis with azoles is strongly recommended with InO, but should not be initiated until at least 24 hours upon completion of InO treatment”.
As requested, a mini-paragraph was added as a comment on the comparative rates of IFDs for T-depleting agents: “Alemtuzumab associates with very high IFD rates, mainly attributed to profound and prolonged T-cell depletion. ATG also displays high IFD rates (but lower than alemtuzumab), because its action affects immune system in a multitude of ways (T-cell depletion, modulation of leukocyte/endothelium interactions, B-cells apoptosis, dys-function of dendritic cells, and induction of regulatory and natural killer T-cells), but not detrimentally. IFD rates related to basiliximab use are also considerable, but significantly lower than the aforementioned agents”. Of course, T-depletion does not lead necessarily to IFDs, because the host defenses against fungi rely on the sophisticated interplay between many factors -discussed in detail in the Introduction section.
Correction in lines 611-613 has been made as suggested. Please note that this child had no hematological malignancy, but was treated with alemtuzumab for aplastic anemia.
We added one more sentence in the immune checkpoint inhibitors’ section stressing the issue of high IFD risk in cases with irAEs and treatment combination with TNF-alpha and/or corticosteroids.
RESPONSE TO REVIEWER #1
Thank you for providing a thorough review and your insightful comments.
Introduction: we added the sentence “Concomitant immunosuppressive therapies usually render us uncapable of defining the exact relative risk for IFDs conferred by each drug” in lines 60-62, so as to underline that it is difficult to ascertain the relative contribution of each chemotherapy agent to IFD occurrence.
As regards anti-CD19 and anti-CD20 blockade we added 3 more sentences to elucidate the immunologic consequences observed in these therapies: “CD19 is known to play a pivotal role in host defense against fungi by interfering with the threshold for B-cell activation, the complement cascade, and major histocompatibility complex (MHC) class II mediated signaling (affecting the interaction with anti-gen-presenting cells). Moreover, depletion of CD19-positive B-cells has been associated with hypogammaglobulinemia and delayed recovery of CD4+ T-helper cells.” […] “Anti-CD20 blockade also affects Th17 cells, which are destined to protect mucosal barriers and contribute to pathogen clearance at mucosal surfaces.” After these addition we feel that the differences between anti-CD20 and anti-CD19 have been adequately described.
The study of Sarlo et al. (2020) had 416 participants and that sample size has been added in text.
The sentence “Delayed antifungal prophylaxis with azoles for 376 at least 24 hours after completion of an InO course is strongly recommended” has been rephrased: “Antifungal prophylaxis with azoles is strongly recommended with InO, but should not be initiated until at least 24 hours upon completion of InO treatment”.
As requested, a mini-paragraph was added as a comment on the comparative rates of IFDs for T-depleting agents: “Alemtuzumab associates with very high IFD rates, mainly attributed to profound and prolonged T-cell depletion. ATG also displays high IFD rates (but lower than alemtuzumab), because its action affects immune system in a multitude of ways (T-cell depletion, modulation of leukocyte/endothelium interactions, B-cells apoptosis, dys-function of dendritic cells, and induction of regulatory and natural killer T-cells), but not detrimentally. IFD rates related to basiliximab use are also considerable, but significantly lower than the aforementioned agents”. Of course, T-depletion does not lead necessarily to IFDs, because the host defenses against fungi rely on the sophisticated interplay between many factors -discussed in detail in the Introduction section.
Correction in lines 611-613 has been made as suggested. Please note that this child had no hematological malignancy, but was treated with alemtuzumab for aplastic anemia.
We added one more sentence in the immune checkpoint inhibitors’ section stressing the issue of high IFD risk in cases with irAEs and treatment combination with TNF-alpha and/or corticosteroids.
Reviewer 2 Report
Brief summary
This review analyzes MEDLINE/PubMed and Cochrane Library for English language publications and peer-reviewed papers regarding invasive fungal diseases (IFD) in pediatric population with hematological malignancies treated with therapy targeting different antigens on the surface of lymphocytes. The review succesfully defined therapies in which IFD are more common complications and suggested antifungal prophylaxis in these cases as well as need for further surveillance.
Broad comments
IFDs in children with hematological malignancies treated with monoclonal antibodies that target surface antigens and immune checkpoint inhibitors are in detail described regarding subjects, concomitant therapy, prophylaxis and outcome. Although in CAR T-cell recipients Table 3. includes only four references with 3/4 referring to early (≤ 28 days) post-CAR-T cells infusion phase, comment on the period after CAR T-cell infusion in which IFD could be expected could be added in the text.
.Specific comments
Data in text and tables are presented in detail, sistematically and appropriately.
Author Response
RESPONSE TO REVIEWER #2
Thank you for your valuable comments. We added the sentence: “Interestingly, another cohort of adults treated with CD19 CAR-T cells for DLBCL dis-played a cumulative incidence of IFD at one year at 4%, stressing the issue for vigilant monitoring for late (>28 days) IFDs” -so as to underline the need for careful monitoring for IFDs beyond the first 28 days after treatment.